# Bioinformatic Identification of CRISPR–Cas Systems in *Leptospira* Genus: An Update on Their Distribution Across 77 Species

**DOI:** 10.3390/pathogens14101044

**Published:** 2025-10-16

**Authors:** Ronald Guillermo Peláez Sánchez, Juanita González Restrepo, Santiago Pineda, Alexandra Milena Cuartas-López, Juliana María Martínez Garro, Marco Torres-Castro, Rodrigo Urrego, Luis Ernesto López-Rojas, Jorge Emilio Salazar Florez, Fernando P. Monroy

**Affiliations:** 1Life and Health Sciences Research Group, Graduate School, CES University, Medellin 050021, Colombia; gonzalez.juanita@uces.edu.co (J.G.R.); pineda.gomez@uces.edu.co (S.P.); milecuartas@gmail.com (A.M.C.-L.); 2CES Biology, Science and Biotechnology School, CES University, Medellin 050021, Colombia; jmartinezg@ces.edu.co; 3Dr. Hideyo Noguchi Regional Research Center, Laboratory of Zoonoses and Other Vector-Borne Diseases, Autonomous University of Yucatan, Mérida 97000, Mexico; antonio.torres@correo.uady.mx; 4INCA-CES Group, School of Veterinary Medicine and Zootechnic, Universidad CES, Medellín 050021, Colombia; rurrego@ces.edu.co; 5Colombian Institute of Tropical Medicine (ICMT), Sabaneta 055450, Colombia; lelopez@ces.edu.co; 6GEINCRO Research Group, School of Health Sciences, San Martin University, Sabaneta 055457, Colombia; jorge.salazarf@sanmartin.edu.co; 7Department of Biological Sciences, Northern Arizona University, Flagstaff, AZ 86011, USA; fernando.monroy@nau.edu

**Keywords:** *Leptospira* species, CAS proteins, leader sequence, spacers, direct repetitions, crRNA, Protospacer Adjacent Motif, genetic editing

## Abstract

Leptospirosis is a globally distributed zoonotic disease caused by pathogenic bacteria of the *Leptospira* genus. Genome editing in *Leptospira* has been difficult to perform. Currently, the functionality of the CRISPR-Cas system has been demonstrated in species such as *Leptospira interrogans*. However, the different CRISPR-Cas systems present in most of the 77 species are unknown. Therefore, the objective of this study was to identify these arrays across the genomes of all described *Leptospira* species using bioinformatics tools. Methods: a bioinformatics workflow was followed: genomes were downloaded from the NCBI database; Cas protein detection was carried out using the CRISPR-CasFinder and RAST web servers; functional analyses of Cas proteins were performed with InterProScan, ProtParam, Swiss Model, Alphafold3, Swiss PDB Viewer, and Pymol; conservation pattern detection was conducted using MEGA12, and Seqlogos; spacer identification was carried out with the Actinobacteriophages database and BLAST version 1.4.0; and bacteriophage detection was performed using PHASTER, and PHASTEST. Results: Cas proteins were detected in 36 out of the 77 species of the *Leptospira* species, including Cas1 to Cas9 and Cas12. These proteins were classified into Class 1 and Class 2 systems, corresponding to types I, II, and V. Direct repeats and spacers were detected in 19 species, with the direct repeats exhibiting two conserved nucleotide motifs. Analysis of spacer sequences revealed 323 distinct bacteriophages. Additionally, three intact bacteriophages were detected in the genomes of four *Leptospira* species. Notably, two saprophytic species have complete CRISPR-Cas systems. Conclusions: The presence of Cas proteins, direct repeats, and spacer sequences with homology to bacteriophage genomes provides evidence for a functional CRISPR-Cas system in at least 19 species.

## 1. Introduction

Leptospirosis is a globally distributed zoonosis disease caused by pathogenic bacteria of the *Leptospira* genus [1]. The disease affects wild and domestic animals, and the bacteria can survive in moist soils and environmental water sources for long periods of time. Moreover, humans are considered accidental hosts of the disease [2]. Leptospirosis is considered a neglected disease that occurs primarily in tropical regions of developing countries [3]. Outbreaks of the disease have been associated with occupations involving animal handling, flooding, environmental disasters, extreme climate change, water sports, prolonged exposure to environmental water sources, and wet soils [4]. Additionally, in endemic regions, severe forms such as Weil’s disease and severe pulmonary hemorrhage syndrome have emerged as the leading cause of fatal cases [5]. Epidemiological studies have estimated that 1.03 million cases and 58,900 deaths occur worldwide each year due to leptospirosis [6]. Currently, the *Leptospira* genus comprises 77 genomic species, which are distributed into pathogenic subgroups (P1 and P2) and saprophytic subgroups (S1 and S2) [7,8,9,10,11,12].

Genome editing in *Leptospira* has been difficult to perform because the bacteria grow slowly in culture media (liquid and solid) and have high nutritional requirements [13]. Gene editing techniques such as random transposon mutagenesis [14,15,16,17], suicide plasmids [18], allelic exchange [19], and shuttle vectors [20] have been used in the *Leptospira* genus. However, these techniques are inefficient, laborious, expensive, and difficult to implement [13]. The lack of efficient gene-editing tools in *Leptospira* has hindered research on hypothetical proteins, pathogenicity mechanisms, virulence factors, and key proteins for drug and vaccine development. Therefore, generating knockout mutant strains remains challenging.

However, the CRISPR-Cas system (Clustered Regularly Interspaced Palindromic Repeats—associated with Cas proteins) has emerged as a novel tool that is highly specific, efficient, easy to design and implement, broadly applicable to bacterial species, and cost-effective [21,22,23]. Additionally, multiple classes, types, subtype, variants, and molecular targets of CRISPR-Cas systems have been described in other bacterial species [24,25,26]. The CRISPR-Cas system is an adaptive immune mechanism in prokaryotes that prevents phage and plasmid infections by storing immunological memory in DNA fragments (spacers). These spacers are integrated into the host chromosome and regularly separated by direct repeats, forming CRISPR arrays adjacent to the leader sequence. These arrays are typically located near genes encoding Cas proteins [27,28,29]. Such proteins are responsible for detecting, cleaving and storing foreign genetic material from bacteriophages or plasmids, which can later serve as molecular guides complementary to the genomic sequences of these invaders during future infections [30,31,32]. The CRISPR-Cas system functions through three main steps: adaptation, expression, and interference. During adaptation (also called insertion or acquisition), foreign genetic material from bacteriophages or plasmids is recognized, cleaved into small fragments, and incorporated in the CRISPR array near the leader sequence. Expression consists of the transcription of the CRISPR locus into precursor RNAs (pre-crRNA), which are subsequently processed into mature CRISPR RNAs (crRNAs). Finally, in the interference step, mature crRNAs associate with multiprotein effector complexes or single effector proteins, which recognize identical or similar sequences in the genomes of invading viruses or plasmids and cleave them to prevent infection [33,34]. This system has been emulated, modified, and adapted as a genome-editing tool in both prokaryotic and eukaryotic cells [35,36]. Therefore, investigating the presence and diversity of these endogenous systems in the genomes of the *Leptospira* genus is of particular interest.

In recent years, significant progress has been made in applying CRISPR-Cas-based gene silencing to the genus *Leptospira*, including the detection of subtypes IB, and IC in pathogenic species [37,38,39]. Additionally, it has been shown that these subtypes are transcriptionally active [40], form interference complexes and different protospacer adjacent motifs (PAMs) have been evaluated to optimize cutting sites [41], and process immature RNAs (Pre-crRNA) [42]. Furthermore, Cas1, Cas2, Cas4, Cas5, and Cas6 proteins have been biologically and functionally characterized [37,38,39,40,42,43,44]. The first attempts at gene silencing were unsuccessful, because the Cas9 protein includes double-strand breaks in DNA that are lethal to *Leptospira* cells [13,45,46]. Therefore, techniques such as CRISPR interference (CRISPRi) [13], CRISPR-Cas9/non-homologous end-joining components (NHEJ) [45], and CRISPR-prime editing [46] have been developed, and successfully used to generate knock-out mutants of multiple proteins in both pathogenic and saprophytic *Leptospira* species. Interestingly, the analysis of the spacer sequences detected in the genomes of *Leptospira interrogans* corresponds to mobile genetic elements, indicating their functionality and importance in the adaptive immune defense of the bacteria [47]. In parallel, applications based on the components of the CRISPR-Cas system have been used as diagnostic tools, including a CRISPR dFnCas9-based quantitative lateral flow immunoassay and CRISPR/Cas12a platform combined with isothermal amplification [48]. Additionally, the nucleotide variability of CRISPR-Cas systems in the *Leptospira* genus has been used as a taxonomic tool for the identification and differentiation of species and serovars in 18 *Leptospira* species [47].

However, little information is available regarding the presence, diversity, and functionality of the CRISPR-Cas machinery, including CAS proteins, spacers, direct repeats, leader sequences, crRNA, and protospacer adjacent motifs (PAMs), across all species of the *Leptospira* genus. Therefore, the objective of this research was to identify the different CRISPR-Cas systems in the 77 species of the *Leptospira* genus. This knowledge may contribute to optimizing gene-editing approaches and to elucidate the biological function of the many proteins that remain hypothetical in *Leptospira*.

## 2. Materials and Methods

### 2.1. Downloading Reference Genomes

A reference genome for each of the 77 *Leptospira* species was downloaded from the NCBI-Genome database (https://www.ncbi.nlm.nih.gov/Taxonomy/Browser/wwwtax.cgi?id=171, accessed on 15 July 2025). Genomes were grouped according to established subgroups (P1, P2, S1, and S2). For each species, the sequences of the two chromosomes (three in some saprophytic species) were concatenated into a continuous linear sequence using the CAT command in the Linux operating system (Ubuntu 24.04 LTS). Genomes were then organized by species subgroups, and the genomic characteristics (species names, clade, accession number, genome size (Mb), GC content (%), number of genes, proteins, and non-coding sequence) were annotated with Prokka v1.14.5 (https://github.com/tseemann/prokka, accessed on 15 July 2025), Pseudopipe (https://github.com/mosilab/pseudopipe, accessed on 15 July 2025), Pseudofinder (https://github.com/filip-husnik/pseudofinder, accessed on 15 July 2025), and RetroScan (https://github.com/Vicky123wzy/RetroScan, accessed on 15 July 2025) [49]. Finally, the genomic characteristics of the different species were compared.

### 2.2. Cas Proteins Detection

The concatenated sequences of each of the 77 species of the *Leptospira* genus were analyzed using the web server RAST (Rapid Annotations using Subsystem Technology) (https://rast.nmpdr.org/rast.cgi, accessed on 15 July 2025) to identify genes encoding Cas proteins, spacers and direct repeats of the CRISPR-Cas system. Each genome was analyzed independently, and the results were consolidated in a database using the Excel program (Microsoft^®^—Microsoft 365). The results were analyzed and grouped into three categories: Cas protein, spacer, or direct repetitions. RAST assigns gene functions and metabolic pathways by comparing annotated genomes; its results are based on subsystems and protein families that are manually curated within the SEED integration, providing the basis for metabolic reconstructions and gene function assignments [50,51,52].

In parallel, the same 77 *Leptospira* species were analyzed using the CRISPRCasFinder web server (https://crisprcas.i2bc.paris-saclay.fr, accessed on 15 July 2025) to detect genes related to the CRISPR-Cas system. As with the RAST analysis, each sequence was examined independently, and the results were consolidated using the Excel program (Microsoft^®^—Microsoft 365). The results were analyzed and grouped into three categories: Cas protein, spacer, or direct repetitions. CRISPRCasFinder program enables the detection of CRISPR arrays and Cas genes in user-supplied sequence data. This is an updated version of CRISPRFinder with increased specificity and improved accuracy in detecting CRISPR-associated sequences. MacSyFinder is used to identify Cas genes, CRISPR-Cas type, and subtype [53].

Finally, the datasets generated by RAST and CRISPRCasFinder were merged into a single database. Duplicate entries were carefully checked and removed, resulting in a non-redundant dataset for downstream comparative analyses.

### 2.3. Analysis of the Biological Function of Cas Proteins

Sequences encoded by CRISPR-Cas-associated genes were identified and downloaded using the BLAST algorithm version 1.4.0 (https://blast.ncbi.nlm.nih.gov/Blast.cgi, accessed on 15 July 2025) from the NCBI database [54].

Functional annotation was performed with InterProScan bioinformatics program (https://www.ebi.ac.uk/interpro/search/sequence/, accessed on 15 July 2025), to identify protein domains, biological processes, molecular function, and cellular components [55]. Additional biological information for each protein was obtained from the Panther GO Terms (https://pantherdb.org/, accessed on 15 July 2025) and GeneCards databases (https://www.genecards.org/, accessed on 15 July 2025). Biological parameters of the proteins were calculated using the ProtParam web server (https://web.expasy.org/protparam/, accessed on 15 July 2025).

The three-dimensional structures of proteins were modeled with SwissModel web server (https://swissmodel.expasy.org, accessed on 15 July 2025) [56], and AlphaFold3 Server (https://alphafoldserver.com/about, accessed on 15 July 2025) [57]. Structural visualization and analysis were carried out with Swiss-pdbViewer version 4.1 (http://spdbv.unil.ch, accessed on 15 July 2025) [58] and PyMOL version 2.6 (https://www.pymol.org/, accessed on 15 July 2025) [59]. 

### 2.4. Classification of Cas Proteins into Functional Stages

The CRISPR-Cas proteins detected were manually classified according to the functional stage in which they participate: adaptation (integration of spacers), expression (processing of pre-crRNA), or interference (participation in the effector complex or target cleavage), based on the currently known biological functions of each protein in the CRISPR-Cas system.

### 2.5. Conservation Patterns of Direct Repeats in the Leptospira Genus

A search for direct repeats was conducted across the 77 species of the *Leptospira* species. The following direct repeats were found according to the subgroups: P1, 332 sequences; P2, 48 sequences; S1, 26 sequences; and S2, 18 sequences. A database was compiled, and multiple sequence alignment was performed per clade using MEGA version 12 (https://www.megasoftware.net/, accessed on 15 July 2025) [60]. The alignments were subsequently visualized with SeqLogos web server (http://imed.med.ucm.es/Tools/seqlogo.html, accessed on 15 July 2025) to determine nucleotide frequencies in the spacer sequences [61].

### 2.6. Bioinformatic Identification of the Immunological Memory (Spacer)

The actinobacteriophage database at phagesDB.org (https://phagesdb.org/, accessed on 15 July 2025) was used to identify the immunological memory of *Leptospira* species against different bacteriophages [62]. For this analysis, spacer sequences detected in the 77 genomes using RAST (Rapid Annotations using Subsystem Technology) and CRISPRCasFinder web servers were used.

### 2.7. Identification of Unique Effector Proteins

The genomes of the 77 *Leptospira* species were scanned using the BLAST algorithm version 1.4.0 (https://blast.ncbi.nlm.nih.gov/Blast.cgi, accessed on 15 July 2025) of the NCBI database to detect the Cas9, Cas12, and Cas13 proteins, which are unique effector proteins in CRISPR-Cas systems [54].

### 2.8. Bioinformatic Detection of Intact Bacteriophages in Genomes

The 77 genomes of the *Leptospira* species were scanned using the PHASTER program (Phage Search Tool Enhanced Release; https://phaster.ca/, accessed on 15 July 2025) and the PHASTEST web server (Phage Search Tool with Enhanced Sequence Translation; https://phastest.ca/, accessed on 15 July 2025). PHASTER is an upgraded version of the PHAST web server that enables rapid identification and annotation of prophage sequences in bacterial genomes and plasmids. PHASTEST further extends this functionality by providing enhanced sequence translation and visualization capabilities for prophage detection [63,64,65].

## 3. Results

### 3.1. Downloading Reference Genomes

In this research, the 77 reference genomes for the *Leptospira* genus were analyzed and classified into the following subgroups of P1 (21 species), P2 (22 species), S1 (29 species) and S2 (5 species). Based on assembly status, 14 genomes were assembled at the chromosome level, 8 at the scaffold level, and 55 at the contig level. Genome sizes ranged from 3.7 to 4.9 with *Leptospira ainazelensis* having the largest genome (4.9 Mb) and *Leptospira fletcheri* and *Leptospira fluminis* the smallest (3.7 Mb). The genomic GC content ranged from 35% present in *Leptospira interrogans* to 48% in *Leptospira ellisii*.

Gene counts ranged from 3409 in *Leptospira fletcheri* and 4789 in *Lepstopira adleri*. Similarly, protein counts varied between 3326 and 4626, respectively, in the same species. The number of pseudogenes ranged from 39 in *Leptospira ryugenii* and *Leptospira ellinghausenii* to 352 in *Leptospira mayottensis* (Table 1).

### 3.2. Cas Proteins Detection

The main objective of this research was to analyze the reference genomes, detect the proteins belonging to the CRISPR-Cas system and classify them according to their class, type, subtype, genetic variants, and molecular target. The presence of these proteins was detected bioinformatically in 36 out of the 77 species of the *Leptospira* genus. The number of species containing Cas proteins in the respective subgroups were as follow: 15 in P1; 11 in P2; 7 in S1; and 3 in S2. From the 36 species that have genes encoding Cas proteins, 19 species had multiple Cas proteins, and 17 species only had the Cas3 protein, with only 3 of them having the Cas3 and Cas3a variants. In terms of class distribution, 32 species were classified as Class 1, and 4 species as Class 2 (*L. gorisiae*, *L. fletcheri*, *L. inadai*, and *L. ilyithenensis*). Notably, *L. gorisiae* and *L. inadai* exhibited proteins from both Classes 1 and 2. Across the genus, proteins from CRISPR-Cas types I, II, III, and V were detected. At the subtype level, proteins belonging to subtypes IA, IB, IE, IC, IIB, and U were detected identified. Variants detected included Cas3a, Cas5a, Cas5a2, Cas5b, Cas5c, Cas7b, Cas7c, Cas8c, and Cas8a1a3.

With respect to molecular targets, 35 species encoded proteins that cleave DNA molecules, while only one species (*L. haakeii*) encoded a protein capable of RNA cleavage (csm2_TypeIIIA). Importantly, this work reports for the first time the presence of CRISPR-Cas system in two saprophytic species (*L. ilyithenensis and L. ryugenii*), both taxonomically assigned to the S2 subgroup (Table 2).

### 3.3. Analysis of the Biological Function of Cas Proteins

Currently, 13 proteins are described as components of the CRISPR-Cas system (Cas1–Cas13). Analysis of the 77 *Leptospira* genomes revealed the presence of 10 out of 13 Cas proteins: Cas1, Cas2, Cas3, Cas4, Cas5, Cas6, Cas7, Cas8, Cas9, and Cas12. In contrast, Cas10, Cas 11, and Cas13 proteins were not detected in any of the genomes analyzed. Furthermore, Cas1 was found in 18 species, Cas2 in 19, Cas3 in 35, Cas4 in 10, Cas 5 in 17, Ca6 in 14, Cas7 in 17, Cas8 in 5, Cas9 in 1, and Cas12 in 3. Cas3 was the most frequent, present in 35 out of the 36 species encoding Cas proteins, while the least frequent were the unique effector proteins Cas 9 and Cas12.

To study the biological function, biological processes, three-dimensional structure, and identify the functional domains of the CAS proteins, the following proteins and species were used as models: Cas1 (*Leptospira interrogans*) is a DNA nuclease and participates in the adaptation stage of the CRISPR-Cas system. It helps in the process of spacer integration into the CRISPR matrix. It is associated with biological processes including defense against bacteriophages, plasmids, as well as the maintenance of repetitive elements within the CRISPR array. The protein consists of 254 amino acids, with an isoelectric point of 9.51, a molecular weight of 29,817.57 Da, a predicted bacterial half-life exceeding 10 h, and an instability index of 43.37, classifying it as unstable. Structurally, it is composed of 8 β-sheets, 18 α-helices, and contains a Cas1_I-II-II domain that may facilitate the linkage of DNA fragments to the CRISPR array (Figure 1).

Cas2 (*Leptospira interrogans*) functions as an RNA nuclease involved in the adaptation stage of the CRISPR-Cas system, contributing to spacer integration into the CRISPR array. It is associated with biological processes such as defense against bacteriophages and plasmids, as well as the maintenance of repetitive elements. The protein consists of 90 amino acids, with an isoelectric point of 8.49, a molecular weight of 10,317.93 Da, a predicted bacterial half-life exceeding 10 h, and an instability index of 31.57, classifying it as stable. Structurally, Cas2 contains 10 β-sheets and 4 α-helices, and includes a SSF:TTP0101/SSO1404-like domain. Cas2 proteins have been described as either endoribonuclease (acting on ssRNA) or endodeoxyribonuclease (acting on dsDNA), depending on the system in which they occur (Figure 1).

Cas3 (*Leptospira interrogans*) is a multifunctional protein with nuclease and helicase activity, participating in the interference stage of the CRISPR-Cas system (target cleavage). It is associated with biological processes such as defense against bacteriophages and plasmids, ATP binding, hydrolase activity, and DNA binding. The protein consists of 377 amino acids, with an isoelectric point of 8.59, a molecular weight of 43,082.83 Da, and a bioinformatically predicted half-life in bacteria of more than 10 h. Its instability index is 37.43, classifying it as stable. Structurally, Cas3 contains 6 β-sheets and 21 α-helices, and exhibits motifs characteristic of superfamily 2 helicases, including a DEAD/DEAH box region and a conserved C-terminal domain. The Cas3-type HD domain confers nuclease activity against ssDNA and ssRNA (Figure 1).

Cas4 (*Leptospira interrogans*) is a DNA nuclease that participates in the adaptation stage of the CRISPR-Cas system (spacer integration). It is linked to biological processes such as defense against bacteriophages and plasmids. The protein consists of 142 amino acids, with an isoelectric point of 8.95, and a molecular weight of 16,606.38 Da, a predicted bacterial half-life exceeding 10 h, and an instability index of 50.59, classifying it as unstable. Structurally, it contains 6 β-sheets and 6 α-helices. The endonuclease activity of Cas4 proteins contributes to crRNA generation by cleaving sequences complementary to foreign genetic elements, and Cas4 has been characterized as a 5′→3′ single-stranded DNA exonuclease (Figure 1).

Cas5 (*Leptospira interrogans*) is a protein that participates in the interference stage of the CRISPR-Cas system (effector complex). It is associated with biological processes including defense against bacteriophages and plasmids, maintenance of CRISPR repeat elements, and RNA binding. The protein consists of 234 amino acids, with an isoelectric point of 7.72, a molecular weight of 27,128.07 Da, a predicted bacterial half-life exceeding 10 h, and an instability index of 46.99, classifying it as unstable. Structurally, it contains 10 β-sheets and 3 α-helices. Cas5 contributes to the processing or stabilization of pre-crRNA into individual crRNA units, and together with Cas6 is required for optimal crRNA maturation and stability (Figure 1).

Cas6 (*Leptospira santarosai*) is a protein that participates in the expression stage of the CRISPR-Cas system (pre-crRNA processing). It is associated with processes such as defense against bacteriophages and plasmids. The protein consists of 203 amino acids, with an isoelectric point of 9.81, a molecular weight of 22,721.75 Da, a predicted bacterial half-life exceeding 10 h, and an instability index of 25.34, classifying it as stable. Structurally, it contains 16 β-sheets and 5 α-helices. Members of this protein family are found across multiple CRISPR-Cas subtypes. Cas6 proteins share the ability to recognize and cleave a single phosphodiester bond with short-repeated sequence of the pre-crRNA transcript (Figure 1).

Cas7 (*Leptospira interrogans*) is a protein that participates in the interference stage of the CRISPR-Cas system (effector complex). It is associated with biological processes such as defense against bacteriophages and plasmids, RNA recognition, and crRNA binding. The protein consists of 279 amino acids, with an isoelectric point of 6.68, a molecular weight of 31,146.15 Da, a predicted bacterial half-life exceeding 10 h, and an instability index of 38.92, classifying it as stable. Structurally, it contains 18 β-sheets and 6 α-helices. Cas7–11 variants can cleave single-stranded RNA (ssRNA) and self-process pre-crRNA (guide RNA). In other systems, Cas7 (DevR) has also been implicated in fruiting body development, sporulation and aggregation (Figure 1).

Cas8 (*Leptospira santarosai*) is a protein participates in the interference stage of the CRISPR-Cas system (effector complex). It is associated with biological processes such as defense against bacteriophages, plasmids, functions as the large subunit of the Cascade complex. The protein consists of 532 amino acids, with an isoelectric point of 9.51, a molecular weight of 61,404.33 Da, a predicted bacterial half-life exceeding 10 h, and an instability index of 31.72, classifying it as stable. Structurally, it contains 23 β-sheets and 19 α-helices. In *Myxococcus xanthus*, the Cas8a1 homolog (also known as DevT) stimulates the synthesis of a signal transduction protein required for fruiting body morphogenesis, a process in which rod-shaped cells differentiate into spherical spores under starvation conditions (Figure 1).

Cas9 (*Leptospira fletcheri*) is a DNA nuclease that participates in the interference stage of the CRISPR-Cas system (target cleavage). It is associated with biological processes such as defense against bacteriophages and plasmids. The protein consists of 1469 amino acids, with an isoelectric point of 9.43, a molecular weight of 17,0367.32 Da, a predicted bacterial half-life exceeding 10 h, and an instability index of 43.69, classifying it as unstable. Structurally, it contains 42 β-sheets and 51 α-helices. Cas9 is inactive in the absence of the two guide RNAs (gRNAs). It recognizes a short sequence motif known as the protospacer adjacent motif (PAM), which enables discrimination between self of non-self DNA, since CRISPR loci lack PAM sequences. PAM recognition is essential for catalytic activity, and cleavage of target DNA occurs only when Cas9 is bound to its gRNAs (Figure 1).

Cas12 (*Leptospira inadai*) is a DNA nuclease and participates in the interference stage of the CRISPR-Cas system (target cleavage). It is associated with biological processes such as defense against bacteriophages and plasmids. The protein consists of 1250 amino acids, with an isoelectric point of 9.26, a molecular weight of 148,356.40 Da, a bioinformatically predicted bacterial half-life of more than 2 min, and an instability index of 30.29, classifying it as stable. Structurally, it contains 33 β-sheets and 50 α-helices. The CRISPR-associated protein Cas12a (Cpf1) possesses two distinct nuclease activities: an endoribonuclease activity that processes its own guide RNAs and an RNA-guided DNase activity that cleaves target DNA. Cas12a, an RNA-guided endonuclease of the type V CRISPR-Cas system adopts a bilobed architecture composed of an α-helical recognition (REC) lobe and a nuclease (NUC) lobe. The crRNA-target DNA heteroduplex binds within the positively charged central channel between the two lobes (Figure 1).

### 3.4. Classification of Cas Proteins into Functional Stages

The CRISPR-Cas system degrades foreign genetic material from viruses through three functional stages: adaptation (adapter integration), expression (pre-crRNA processing), and interference (effector complex formation and target cleavage). Among the 36 species with genes encoding Cas proteins, 19 encoded multiple Cas proteins, whereas 17 encoded only Cas3 protein. The latter group likely represents species with a partial or incomplete CRISPR-Cas system.Therefore, the purpose of this analysis was to establish the distribution of Cas proteins across the three functional stages of the CRISPR-Cas system in the 19 *Leptospira* species that encode multiple proteins, and to identify species with a complete system. Fifteen species carried Class 1, Type I CRISPR-Cas system, one species (*L. fletcheri*) had a Class 2, Type II system, and one species (*L*. *ilyithenensis*) had a Class 2, Type V system. Two species, *L. gorisiae* and *L. inadai*, encoded a combination of Class 1, Type I and Class 2, Type V systems.

Notably, *L. kirschneri*, *L. interrogans*, and *L. noguchii* lacked Cas6, which is normally required for the expression process (pre-crRNA processing), although Cas5 may compensate for this function. The absence of Cas1 in *L. borgpetersenii*, and Cas8 in *L. adleri*, *L. alstonii*, and *L. inadai*, was also unexpected. Additionally, the presence of csm2, a protein of the class 1—Type IIIA system in *L. haakeii*, is striking (Table 3).

### 3.5. Conservation Patterns of the Direct Repeats in the Leptospira Genus

A search for direct repeats was performed across the 77 species, identifying 434 spacer sequences distributed among the four species subgroups. Spacer length ranged from 23 to 39 nucleotides. Alignment of these sequences and analysis of nucleotide frequency reveled subgroup-specific conservation patterns: in P1, a central conserved motif (GTTTGAACTCCCACAAGTT), in P2 a central motif (CTGATCCCCACACACGTGGGGATTAA), in S1 a central motif (CCAGATCTGCAAGTGGATCTGC), and in S2, conservation was observed across the entire sequence (CTCACCACGCATATGGGGTTCAACCAT). Furthermore, when the 434 sequences from all four subgroups were compared collectively, no conservation pattern was detected (Figure 2).

### 3.6. Bioinformatic Identification of the Immunological Memory (Spacer Sequences)

This analysis is performed on the 19 species with the presence of spacer sequences, to verify the functionality of the different CRISPR-Cas systems detected. The presence of spacer sequences within the CRISPR arrays (Leader Sequence—Direct Repeats—Spacer Sequences) is clear evidence of the functionality of the different systems, because the bacteria incorporate spacer sequences as a kind of immunological memory against bacteriophages or plasmids. Among the 19 *Leptospira* species with Cas proteins, all presented spacer sequences and direct repeats. Among the 19 species with a complete and functional CRISPR-Cas system (presence of Cas proteins, spacer sequences, and direct repeats), a range of (12–58) spacers per species was detected. *Leptospira noguchii* with 58 spacer sequences was the species with the widest repertoire of immunological memory against bacteriophages, while *Leptospira kirschneri* with 12 spacer sequences was the species with the lowest immunological memory. A total of 617 spacer sequences were detected among the 19 *Leptospira* species, but only 323 were identified as belonging to different bacteriophages in the database (The actinobacteriophage database). Forty-two spacer sequences could not be identified, representing bacteriophages not reported in the database. These spacer sequences that represent the different bacteriophages have been found in the genome of 12 bacterial genera (*Gordonia*, *Arthrobacter*, *Streptomyces*, *Rhodococcus*, *Propionibacterium*, *Microbacterium*, *Mycobacterium*, *Curtobacterium*, *Brevibacterium*, *Corynebacterium*, and *Tsukamurella*). It should be noted that these 12 bacterial genera are represented by multiple spacer sequences that were found in various *Leptospira* species (Appendix A).

### 3.7. Identification of Unique Effector Proteins

Given the difficulty of establishing gene-editing systems with CRISPR-Cas complexes composed of multiple effector proteins, a search was conducted for single-protein effectors such as Cas9, Cas12, and Cas13. One species (*L. fletcheri*) encoded Cas9, a protein Class 2, Type II, while three species (*L. ilyithenensis*, *L. gorisiae*, and *L. inadai*) encoded Cas12, a protein Class 2, Type V. No species were found with Cas13. Cas9 and Cas12 were structurally modeled, and functional domain analyses were performed to better understand the structure and function of these endogenous proteins, which could potentially be used as gene-editing tools in *Leptospira* (Figure 1). When comparing the *Cas9* and *Cas12a* proteins of the *Leptospira* genus with homologs from model organisms commonly used in CRISPR–Cas studies (*Streptococcus pyogenes* Cas9 and *Francisella novicida* Cas12a), we found that both proteins show strong similarity to those identified in other *Leptospira* species. Moreover, they exhibit high sequence identity, genetic relatedness, and coverage with members of phylogenetically related genera such as *Turnerella* and *Leptonema*. Notably, the *Leptospira fletcheri* Cas9 protein differs from *S. pyogenes* Cas9 in amino acid sequence, functional domain composition, and predicted tertiary structure. In contrast, the *Leptospira inadai* Cas12a protein displays nearly identical sequence, domain architecture, and structural configuration to that of *F. novicida*, a well-established model species for CRISPR–Cas research (Appendix A).

### 3.8. Bioinformatic Detection of Intact Bacteriophages in Genomes

Bioinformatic detection of bacteriophages was performed across the genomes of 77 *Leptospira* species. Three intact bacteriophages were identified in four *Leptospira species* (*L. mayottensis*, *L. weilii*, *L. bandrabouensis*, and *L. ellinghausenii*), while two questionable or nearly complete bacteriophages were detected in *L. weilii* and *L. fluminis*. In addition, 58 species contained incomplete bacteriophages, and 13 showed no evidence of bacteriophage sequences (Figure 3).

## 4. Discussion

The *Leptospira* genus has made great progress in recent years, with 77 species de-scribed and 1157 genomes sequenced (https://www.ncbi.nlm.nih.gov/datasets/genome/?taxon=171, accessed on 5 August 2025). Each species has a reference genome and multiple genomes from different strains. These genomes are a valuable source of genetic information for computational biology studies, where different bacterial biological processes can be explored. The major problem with the *Leptospira* genomes, and many prokaryotes, is that a large portion of the proteins encoded in their genomes are hypothetical (predicted by an ORF, but without experimental validation), thus rendering their functions unknown. For example, in the species *L. interrogans* serovar Lai, only 11.5% of the proteins have been characterized [66,67]. According to the functional annotation of the proteins in the 77 genomes of the *Leptospira* genus, we found a range of hypothetical proteins between (41.4–57.9%), being the species with the most hypothetical proteins *L. ainlahdjerensis* (57.9%) and the species with the least hypothetical proteins *L. Harrisiae* (41.4%). This result reflects a large gap in the understanding of biological functions of proteins in the *Leptospira* genus and the urgent need for a gene editing tool (Figure 4).

Moreover, the gene editing process in the *Leptospira* genus has been difficult to standardize, which has hampered the functional analysis of thousands of proteins and their assignment to the bacteria’s biological processes [13]. However, the CRISPR-Cas methodology is emerging as a promising tool for gene silencing in *Leptospira*, yielding knockout mutants of multiple proteins [13,45,46]. Therefore, it is important to explore endogenous CRISPR-Cas systems in the 77 currently described species to uncover their diversity, characterize their molecular components (Cas proteins, direct repeats, and spacers), assess their immunological memory, and identify intact bacteriophages that have evaded the immune system. Regarding genome availability, a reference genome was obtained for each of the 77 *Leptospira* species, providing a comprehensive overview of Cas protein distribution in the genus (Table 1). Similar work has identified CRISPR-Cas systems in 18 *Leptospira* species, with the objective of assessing the taxonomic utility of CRISPR-Cas arrays for distinguishing species and serovars of the genus [47].

A total number of 166 Cas proteins were detected in 36 from the 77 (46.75%) *Leptospira* species, corresponding to Cas1 to Cas9, and Cas12. The proteins were classified into class 1 and class 2 systems, and types I, II, and V. Additionally, proteins belonging to the subtypes IA, IB, IC, IE, IIB, and IIIA, and variants Cas3a, Cas5a, Cas5a2, Cas5b, Cas5c, Cas7b, Cas7c, Cas8a1a3, and Cas8c were also detected. All proteins target DNA for cleavage, except for the csm2_TypeIIIA protein in *L. haakeii*, which also targets RNA molecules. The Csm complex comprises five Cas proteins (Csm1-Csm5) and a crRNA, which degrades invading DNA and RNA [68].

Currently, there are 43 pathogenic species divided into P1 (21 species) and P2 (22 species) subgroups, from which 27 species encoded Cas proteins; therefore, concluding that not all pathogenic species have CRISPR-Cas systems. Moreover, there are 34 saprophytic species grouped into S1 (29 species) and S2 (5) subgroups, with only two species (*L. ilyithenensis*, and *L. ryugenii*) having complete CRISPR-Cas systems and the Cas3a, type I protein present in ten species. Based on these findings, we can conclude that some saprophytic species encode CRISPR-Cas systems, contrary to what was reported in a previous study, that did not include these species [47]. Having found such systems potentially allows for experimental trials without the risk of infection in research laboratories. It is striking to find the cas3a, typeI protein as the sole representative of the CRISPR-Cas system in 17 *Leptospira* species (9 pathogens and 8 saprophytes). This suggests that these species have lost their CRISPR–Cas systems, and this vestigial protein has been conserved in their genomes to perform different functions in other biological processes; beyond defense against viruses (CRISPR immunity), the Cas3 protein may be involved in regulating biofilm formation and virulence in bacteria [69], as well as influencing genes related to quorum sensing (a form of bacterial communication where cells release and detect signal molecules called autoinducers, allowing them to sense the population density of their community and coordinate gene expression to alter collective behavior) [70], secretion systems [71], and flagella formation [69]. These non-canonical functions of Cas3 are being investigated and may have implications for gene editing and other cellular biological processes.

According to the results obtained, the *Leptospira* genus has a wide diversity of Cas proteins in 36 species that could be used as genetic editing tools.

Thus far, 13 types of Cas proteins have been described. Analysis of *Leptospira* genomes, revealed 10 of these proteins 13 (76.92%), indicating that the genus possesses a broad repertoire of Cas components. Eight belong to multi-protein effector systems (Cas1-Cas8), while two correspond to single-protein effectors (Cas9 and Cas12a). Notably, this study reports for the first time the presence of endogenous Cas9 and Cas12a in the *Leptospira* genus.

Previous studies have reported gene silencing in *Leptospira* species using Cas9, but the proteins employed were heterologous, genetically modified nucleases designed to introduce a single cut in the target DNA [46].

A functional bioinformatics analysis for the ten types of Cas proteins detected (Cas1-Cas9, and Cas12a) was performed to characterize their physicochemical properties, biological functions, associated biological processes, three-dimensional structures, and functional domains. This analysis enabled us to assess their roles within the CRISPR-Cas system and to assign each protein to one of the three functional stages: adaptation, expression, or interference. As a result of the analysis, it was found that the functionality of Cas proteins in *Leptospira* genus corresponds to other bacterial species (Figure 1).

Another important step in characterizing the CRISPR-Cas system is the classification of proteins into functional stages according to their biological roles. The first stage, adaptation, involves recognition of foreign DNA and RNA, cleavage of the viral genome, and integration of genome fragments into the bacterial chromosome as spacer sequences. This stage typically includes Cas1, Cas2, and Cas4, although Cas4 may be absent in most systems and entirely lacking in types III, V, and VI [22]. Among the 19 *Leptospira* species encoding Cas proteins, ten contained Cas1, Cas2, and Cas4; 8 contained Cas1 and Cas2; and 1 contained only Cas2. According to our results, the adaptation stage is complete in all 19 species. Although *L. borgpetersenii* lacks Cas2, its system appears to be functional, as spacer sequences were identified originating from bacteriophages.

The functional stage of expression consists of the transcription of the CRISPR microarray locus into immature RNA precursors (pre-crRNA) and their subsequent processing into mature RNA (cr-RNA) [22]. In this state, Cas6 or Cas5 are key components in Class 1, type I system, whereas RNAse III fulfills this role in Class 2, type II system. Additionally, Class 2, types V and VI systems, the unique effector proteins Cas12 and Cas13 also participate in pre-crRNA processing [22]. According to our results, all 19 *Leptospira* species analyzed possess the proteins necessary for a fully functional expression stage. (Table 3).

The interference stage consists of binding mature RNAs (crRNA) to multiprotein effector complexes or single effector proteins, which recognize an identical or similar sequence in the genome of invading viruses or plasmids and cleave them to inactivate them [33,34]. In Class 1 type I systems, Cas5, Cas7, and Cas8 are important to form the effector complex, along with Cas3 protein, which is responsible for targeting and cleaving DNA, whereas in Class 2 type II, V, and VI systems, Cas9, Cas12, and Cas13 are essential to form the effector complex. According to our results, all species have their respective complete effector complexes (Table 3).

Once we verified that the adaptation, expression, and interference stages were complete, we proceeded to assign the classes, types, and subtypes according to the taxonomic classification of CRISPR-Cas systems proposed by Makarova et al. in 2020 [36]. In our study, we found seventeen species classified as Class 1 type I; one species as Class 2 type II; and three species as class 2 type V. The following CRISPR-Cas subtypes were identified: IB (*L. santarosai*, and *L. broomii*), IC (*L. kirschneri*, *L. interrogans*, and *L. noguchii*), IE (*L. alexanderi*, *L. borgpetersenii*, *L. mayottensis*, *L. stimsonii*, *L. weilii*, *L. gorisiae*, *L. fainei*, *L. koniambonensis*, and *L. ryugenii*), IIB (*L. fletcheri*), VA (*L. gorisiae*, *L. inadai*, and *L. ilyithenensis*), IF2 (*L. adleri*, *L. alstonii*, and *L. inadai*). It is worth highlighting that the species *L. inadai* (Class 1 type IF2, and Class 2 type V) and *L. gorisiae* (Class 1, type IE, and Class 2 type V) each contain two distinct CRISPR-Cas systems (Table 3). To date, only subtypes IB, and IC have been described in the *Leptospira* genus [37,38,39]. This study therefore provides the first report of subtypes IE, IIB, VA, and IF2 in *Leptospira*. These findings significantly contribute to our understanding of the diversity and functionality of CRISPR–Cas systems in the *Leptospira* genus. Bacterial species can harbor and activate multiple types of CRISPR–Cas systems, allowing them to defend against a wider range of viral and genetic threats. The presence of multiple systems reflects an evolutionary war with phages, which in turn have developed inhibitors such as anti-CRISPR proteins to evade detection [72]. In addition, an additional system can act as a backup in case the primary CRISPR–Cas system fails, ensuring the continuity of bacterial defense. The presence of multiple CRISPR–Cas systems in a single bacterial species is a known phenomenon: some Streptococcus species can harbor up to three different types of CRISPR–Cas systems [73]. Clostridium and Bacillus genera have multiple types of CRISPR–Cas systems in different strains, such as types IB, II-B, II-C, and III-B in Clostridium [74], and types IB and IC in Bacillus [75].

Another fundamental component of the CRISPR-Cas system is the direct repeats (DR), which serve to separate the spacer sequences (segments of DNA from viruses). These nucleotide sequences are identical in size and sequence. Direct repeats can be similar between related species, but also very different between distant species. The average spacer size is 32 nucleotides, but they can vary between 21 and 47 base pairs [22]. Analysis of the 77 genomes uncovered 434 unique direct repeats, with sizes ranging between 23 and 55 base pairs. Conservation patterns were observed within the species subgroups (P1, P2, S1, and S2), but no conserved pattern was found among the 77 species. This is consistent with the literature, where genetically closely related species retain similar direct repeats, while genetically distant species use completely different direct repeats in their CRISPR-Cas arrays [22] (Figure 2).

Spacers are nucleotide sequences of fixed length but high variability, as they originate from diverse bacteriophages and plasmids. Sizes range from 20 to 72 base pairs [22]. A total of 617 spacer sequences were identified, representing the bacterial immunological memory against bacteriophage infection. Of these, 404 sequences matched entries in the Actinobacteriophage database and corresponded to 323 unique becteriophages. The remaining 81 sequences could not be identified, reflecting the presence of many bacteriophages yet to be reported in the database. The identified bacteriophages were associated with 11 different bacterial genera (Appendix A). These findings confirm the presence and functionality of CRISPR-Cas systems in 19 *Leptospira* species. Structurally, genes encoding Cas proteins, direct repeats, and spacer sequences were detected. Functionally, spacers corresponded to bacteriophages sequences, indicating that *Leptospira* species had been exposed to these bacteriophages, avoided infection, and generated an immunological memory against them. However, the bioinformatics analysis of CRISPR–Cas systems in the *Leptospira* genus is a preliminary result that demonstrates the existence of the components, but it is necessary to experimentally validate these results to ensure that the systems work correctly. For example, this could involve performing experimental evaluations of the degradation of a bacteriophage or blocking a gene in a *Leptospira* species that has the complete CRISPR–Cas system. These findings have significant implications for potential applications in bacteriophage therapy, where the CRISPR–Cas system could be harnessed to develop targeted treatments for bacterial infections.

An interesting result was the partial or total loss of the CRISPR–Cas system in 58 species of the *Leptospira* genus. This raises the following question: why are some *Leptospira* species losing this defense system? Regarding this phenomenon, it is known that some bacterial species lack CRISPR–Cas systems due to evolutionary inactivation. CRISPR–Cas systems can be selectively inactivated in certain bacteria, as observed in the *Bacillus cereus* group [76], allowing them to adapt to specific environments. Another cause of CRISPR–Cas system loss is selective pressure induced by antibiotic exposure. Intensive antibiotic use can lead to the loss of the CRISPR–Cas system, as it can act as a barrier to the acquisition of antibiotic resistance genes [77]. Additionally, total or partial loss of essential components of the CRISPR–Cas system can render the system inoperable. For these reasons, we assume that some species of the *Leptospira* genus have lost their CRISPR–Cas defense system.

CRISPR-Cas systems can cleave sequences from bacteriophages or plasmids using multi-protein complexes (Class 1-types I, III, and IV) or single effector proteins (Class 2-types II, V, and VI). Single effector systems are particularly attractive for gene editing due to their simplicity, which facilitates application across multiple bacterial species [22]. According to our results, three species encode single-protein effector: *L. fletcheri* with a Class 2, type II system (Cas9 as cleavage protein), and *L. inadai* and *L. ilyithenensis* with Class 2, type V systems (Cas12a, as cleavage protein). This finding is significant for the *Leptospira* genus, as it represents the first report of endogenous Cas9 and Cas12a effectors. These proteins could serve as tools to optimize future gene-editing approaches while avoiding the toxicity associated with exogenous proteins from other bacterial species.

Finally, CRISPR-Cas systems are not infallible. Therefore, it is important to identify bacteriophages that have successfully evaded the CRISPR-Cas defense system, leaving their intact sequences embedded into the bacterial genome. Detecting such prophages is crucial not only for understanding the evolutionary dynamics between bacteria and their viruses but also for developing novel biotechnological applications. Bacteriophages have gained attention as alternative tools for gene editing [78,79,80] and as promising candidates for antibacterial therapy against multidrug-resistant infections in humans and animals [81,82,83,84,85].

Based on our analysis, we detected the following bacteriophages infecting four *Leptospira* species: *L. mayottensis* (PHAGE_Pseudo_phi3_NC_030940), *L. weilii* (PHAGE_Paenib_Tripp_NC_028930), *L. bandrabouensis* and *L. ellinghausenii* (PHAGE_Leptos_LE1_NC_048892). The only phages that have been isolated, purified, and phenotypically characterized in the *Leptospira* genus are LE1, LE3, and LE4. These bacteriophages were described infecting the saprophytic species *Leptospira biflexa* in the 1990s [86], and the LE1 genome was subsequently sequenced [87], and in 2018, sequencing and proteomic analyses of LE3 and LE4 were reported [88]. Despite these advances, research on Leptospira bacteriophages remains limited.

In this study, two additional bacteriophages infecting *Leptospira* were identified (PHAGE_Pseudo_phi3_NC_030940, and PHAGE_Paenib_Tripp_NC_028930). These findings expand current knowledge of phage diversity in this genus and may prove valuable for future applications in *Leptospira* genome engineering and the development of next-generation bacteriophage-based antibacterial therapies.

## 5. Conclusions

The presence of Cas proteins, direct repeats, and spacer sequences homologous to bacteriophage genomes indicates a functional CRISPR-Cas system in 19 *Leptospira* species. Notably, the detection of endogenous Cas9 and Cas12a proteins, both unique effectors, highlight their potential as precise tools for gene editing in this genus. The identification of saprophytic species with complete and functional CRISPR-Cas systems suggests a recent acquisition of this system in subgroup S2 and provides non-pathogenic models that may be valuable for experimental studies under laboratory conditions. Additionally, the three intact bacteriophages detected in four *Leptospira* species represent promising candidates both for genetic manipulation of the bacteria, due to their ability to evade the CRISPR-Cas system, and for the development of phage-based therapeutics aimed at selectively lysing *Leptospira* during human and animal infections.

## Figures and Tables

**Figure 1 pathogens-14-01044-f001:**
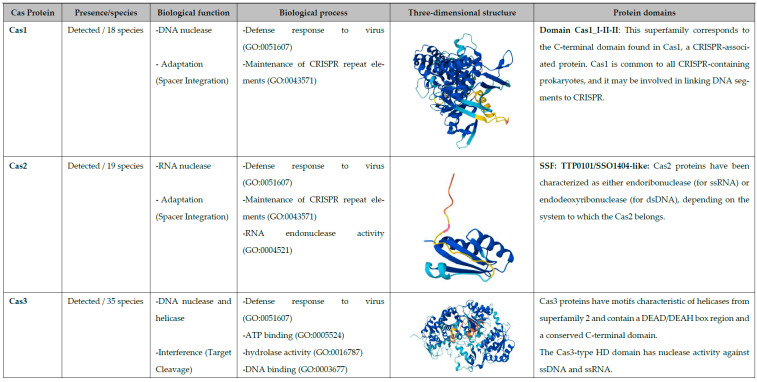
The figure shows the ten Cas proteins described to date and indicates which were detected in the genomes of the 77 species of the *Leptospira* species. Their biological functions and processes were annotated using InterPro. The three-dimensional structures of Cas1, Cas2, Cas3, Cas4, Cas5, Cas6, Cas7, Cas8, Cas9, and Cas12 were modeled with Swiss-Model and AlphaFold3. Functional domains were also identified, and their corresponding biological roles were assigned.

**Figure 2 pathogens-14-01044-f002:**
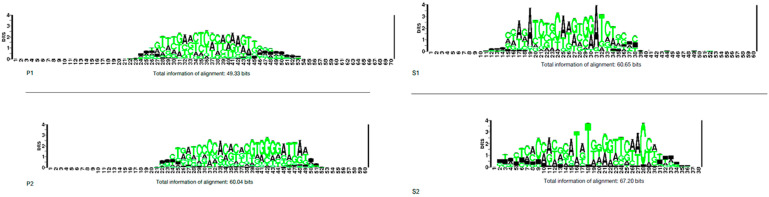
Conservation patterns of direct repeats in the *Leptospira* genus. The figure shows the alignment of spacer sequences (nucleotides) from 30 *Leptospira* species (those encoding Cas proteins, direct repeats, and spacers) and the resulting conservation patterns analysis. Colors in the alignment represent nucleotides: adenine (black), thymine, guanine, and cytosine (green). The size of the letters in the sequence logos is proportional to the nucleotide frequency in the alignment (measured in bits).

**Figure 3 pathogens-14-01044-f003:**
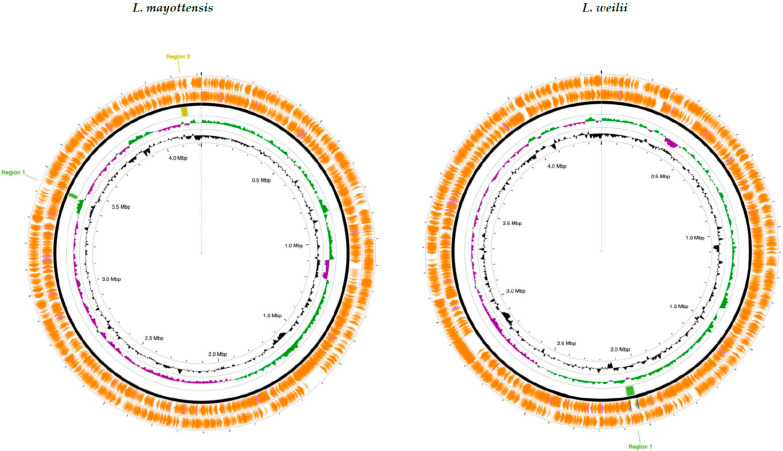
Detection of bacteriophages integrated into the genome of four species of the genus *Leptospira* (*L. mayottensis*, *L. weilii*, *L. bandrabouensis*, and *L. ellinghausenii*), indicating successful evasion of the CRISPR-Cas system. The figure displays the genomic regions where bacteriophage proteins were detected (highlighted in green), including species, genomic location, region length, completeness, score, number of phage proteins identified, insertion site, phage identity, and GC content. Color coding denotes completeness: dark green, intact bacteriophages (complete genome detected); light green, nearly complete bacteriophages (almost the entire genome was detected); red, incomplete bacteriophages (small genomic region detected).

**Figure 4 pathogens-14-01044-f004:**
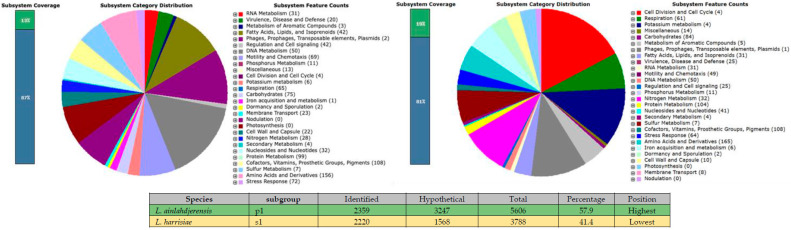
The figure shows the genomic annotation of the *Leptospira* species with the highest and lowest number of hypothetical proteins, determined using the RAST (Rapid Annotation using subsystem Technology) web server. Displayed information includes the species name, subgroup, the annotated or identified proteins, hypothetical proteins, total number of proteins, percentage of hypothetical proteins, and the rank based on hypothetical protein abundance.

**Table 1 pathogens-14-01044-t001:** The table summarizes the 77 *Leptospira* species used in the study, their taxonomic distribution within subgroups (P1, P2, S1, and S2), NCBI Taxonomy accession numbers, genome size (Mb), CG content (%), number of genes, proteins, and non-coding sequences. Color coding indicates subgroup: green for pathogenic P1, blue for pathogenic P2, yellow for saprophytic S1, and red for saprophytic S2.

Species	Subgroup	Access Number	Sequencing Level	Genome Size (Mb)	GC (%)	Genes	Proteins	Non-Coding
*L. adleri*	P1	GCA_002811985.1	Scaffold	4.8	43.5	4789	4626	163
*L. ainazelensis*	P1	GCA_016918785.1	Contig	4.9	42.5	4417	4310	107
*L. ainlahdjerensis*	P1	GCA_016919175.1	Contig	4.8	42.5	4409	4310	99
*L. alexanderi*	P1	GCA_000243815.3	Contig	4.2	40	4582	4541	41
*L. alstonii*	P1	GCA_000347175.1	Contig	4.4	42.5	4423	4380	43
*L. barantonii*	P1	GCA_002811925.1	Contig	4.4	44	4135	4033	102
*L. borgpetersenii*	P1	GCA_003516145.1	Chromosome	4	40	3792	3463	329
*L. ellisii*	P1	GCA_002811955.2	Contig	4.3	48	3998	3896	102
*L. gomenensis*	P1	GCA_004770155.1	Contig	4.3	46	3931	3802	129
*L. gorisiae*	P1	GCA_040833975.1	Chromosome	4.5	41.5	4104	3939	165
*L. interrogans*	P1	GCA_002073495.2	Chromosome	4.6	35	4049	3772	277
*L. kirschneri*	P1	GCA_000243695.3	Contig	4.4	36	4029	3986	43
*L. kmetyi*	P1	GCA_003722295.1	Chromosome	4.4	45	4160	4035	125
*L. mayottensis*	P1	GCA_000306675.3	Chromosome	4.2	39.5	3944	3592	352
*L. noguchii*	P1	GCA_000306255.2	Contig	4.7	35.5	4565	4520	45
*L. sanjuanensis*	P1	GCA_022267325.1	Contig	4.5	45	4152	4052	100
*L. santarosai*	P1	GCA_000313175.2	Chromosome	4	42	4191	4080	111
*L. stimsonii*	P1	GCA_003545875.1	Contig	4.7	42.5	4747	4599	148
*L. tipperaryensis*	P1	GCA_001729245.1	Chromosome	4.6	42.5	4342	4255	87
*L. weilii*	P1	GCA_006874765.1	Chromosome	4.4	41	4312	3965	347
*L. yasudae*	P1	GCA_003545925.1	Contig	4.4	45.5	4286	4162	124
*L. andrefontaineae*	P2	GCA_004770105.1	Contig	4.3	40	3969	3894	75
*L. broomii*	P2	GCA_000243715.3	Contig	4.4	43	4249	4205	44
*L. dzoumogneensis*	P2	GCA_004770895.1	Contig	4.1	41	3814	3724	90
*L. fainei*	P2	GCA_000306235.2	Contig	4.3	43.5	4157	4113	44
*L. fletcheri*	P2	GCA_004769195.1	Contig	3.7	47.5	3409	3326	83
*L. fluminis*	P2	GCA_004771275.1	Contig	3.7	47.5	3427	3342	85
*L. haakeii*	P2	GCA_002812225.1	Scaffold	4.2	40	3920	3814	106
*L. hartskeerlii*	P2	GCA_002811475.1	Scaffold	4.1	40.5	3787	3708	79
*L. inadai*	P2	GCA_000243675.3	Contig	4.5	44.5	4314	4264	50
*L. johnsonii*	P2	GCA_003112675.1	Contig	4.1	41.5	3754	3713	41
*L. koniambonensis*	P2	GCA_004769555.1	Contig	4.3	39	4000	3929	71
*L. langatensis*	P2	GCA_004770615.1	Contig	4.1	45	3751	3671	80
*L. licerasiae*	P2	GCA_000526875.1	Contig	4.2	41	3899	3834	65
*L. neocaledonica*	P2	GCA_002812205.1	Scaffold	4.2	40	3978	3889	89
*L. perolatii*	P2	GCA_002811875.1	Contig	4	42.5	3712	3631	81
*L. saintgironsiae*	P2	GCA_002811765.1	Contig	4.1	39	3830	3736	94
*L. sarikeiensis*	P2	GCA_004769615.1	Contig	4.4	40.5	4026	3928	98
*L. selangorensis*	P2	GCA_004769405.1	Contig	4.2	40	3894	3814	80
*L. semungkisensis*	P2	GCA_004770055.1	Contig	3.9	43	3626	3562	64
*L. venezuelensis*	P2	GCA_002150035.1	Contig	4.3	39	4030	3973	57
*L. wolffii*	P2	GCA_004770635.1	Contig	4.2	46	3851	3771	80
*L. cinconiae*	P2	GCA_040833995.1	Chromosome	4.1	42	3801	3741	60
*L. abararensis*	S1	GCA_016918735.1	Contig	4.2	39	3968	3900	68
*L. bandrabouensis*	S1	GCA_004770555.1	Contig	4.2	37.5	3928	3857	71
*L. biflexa*	S1	GCA_000017685.1	Chromosome	4	39	3775	3726	49
*L. bourretii*	S1	GCA_004770145.1	Contig	4.2	38	3923	3840	83
*L. bouyouniensis*	S1	GCA_004770625.1	Contig	4.1	37	3833	3746	87
*L. brenneri*	S1	GCA_004769295.1	Contig	3.9	38.5	3662	3593	69
*L. chreensis*	S1	GCA_016919165.1	Contig	4.5	40	4176	4086	90
*L. congkakensis*	S1	GCA_004770265.1	Contig	4	38	3712	3647	65
*L. ellinghausenii*	S1	GCA_003114815.1	Contig	4.2	37.5	3960	3921	39
*L. harrisiae*	S1	GCA_002811945.1	Scaffold	3.9	38	3726	3651	75
*L. jelokensis*	S1	GCA_004769775.1	Contig	4.1	39	3833	3755	78
*L. kanakyensis*	S1	GCA_004769235.1	Contig	4.1	38.5	3884	3810	74
*L. kemamanensis*	S1	GCA_004769665.1	Contig	3.8	39	3533	3436	97
*L. levettii*	S1	GCA_002812085.1	Scaffold	3.9	37.5	3655	3579	76
*L. meyeri*	S1	GCA_004368965.1	Contig	4.2	38	4028	3930	98
*L. montravelensis*	S1	GCA_004770045.1	Contig	4	37.5	3760	3686	74
*L. mtsangambouensis*	S1	GCA_004770475.1	Contig	4.1	38	3836	3764	72
*L. noumeaensis*	S1	GCA_004770765.1	Contig	4.1	38.5	3832	3758	74
*L. perdikensis*	S1	GCA_004769575.1	Contig	4	38.5	3740	3680	60
*L. terpstrae*	S1	GCA_000332495.2	Contig	4.1	38	3932	3889	43
*L. vanthielii*	S1	GCA_004770365.1	Contig	4.1	39	3839	3753	86
*L. wolbachii*	S1	GCA_000332515.2	Contig	4.1	39	3956	3912	44
*L. yanagawae*	S1	GCA_004769275.1	Contig	4	38.5	3704	3631	73
*L. mgodei*	S1	GCA_040833985.1	Chromosome	4	39	3807	3752	55
*L. milleri*	S1	GCA_040833955.1	Chromosome	3.9	38.5	3628	3555	73
*L. iowaensis*	S1	GCA_040833965.1	Chromosome	4.1	37	3867	3812	55
*L. paudalimensis*	S1	GCA_026151345.1	Contig	4.1	37.5	3769	3711	58
*L. soteropolitanensis*	S1	GCA_026151335.1	Contig	4.1	37.5	3934	3863	71
*L. limi*	S1	GCA_026151395.1	Contig	3.9	37.5	3679	3619	60
*L. idonii*	S2	GCA_004770995.1	Contig	4.1	41	3797	3724	73
*L. ilyithenensis*	S2	GCA_004771005.1	Contig	4.2	40.5	3950	3849	101
*L. kobayashii*	S2	GCA_003114835.3	Chromosome	4.3	40.5	3945	3902	43
*L. ognonensis*	S2	GCA_004770745.1	Scaffold	4	39.5	3733	3660	73
*L. ryugenii*	S2	GCA_003114855.1	Scaffold	4	40	3698	3659	39

**Table 2 pathogens-14-01044-t002:** The table summarizes the 77 *Leptospira* species analyzed, including their taxonomic subgroup distribution, CRISPR-Cas proteins detected, and their classification into classes, types, subgroups, and variants. The molecular targets cleaved by effector Cas proteins are also indicated. Color coding indicates subgroup: green for pathogenic P1, blue for pathogenic P2, yellow for saprophytic S1, and red for saprophytic S2.

Species	Clade	Proteins	Class	Type	Subtype	Variant	Native Target
*L. adleri*	P1	cas1_TypeIA, cas4_TypeI-II, cas3_TypeI, cas3a_TypeI, cas2_TypeI-II-III, cas5a2_TypeIA, cas6_TypeIA, cas7b_TypeIB	Class 1	I, II, III	IA, IB	cas3a, cas5a2, cas7b	DNA
*L. ainazelensis*	P1	Not detected	-	-	-	-	-
*L. ainlahdjerensis*	P1	cas3_TypeI	Class 1	I	-	-	DNA
*L. alexanderi*	P1	cse1_TypeIE, cse2_TypeIE, cas1_TypeIE, cas2_TypeIE, cas3_TypeI, cas4_TypeI-II, cas5_TypeIE, cas6_TypeIE, cas7_TypeIE	Class 1	I, II	IE	-	DNA
*L. alstonii*	P1	cas1_TypeIA, cas2_TypeI-II-III, cas3_TypeI, cas5a2_TypeIA, cas6_TypeIE, cas7_TypeI, cas4_TypeI-II, cas5_TypeIE	Class 1	I, II, III	IA, IE	Cas5a	DNA
*L. barantonii*	P1	cas3_TypeI	Class 1	I	-	-	DNA
*L. borgpetersenii*	P1	cse2_TypeIE, cse1_TypeIE, cas1_TypeIE, cas2_TypeIE, cas3_TypeI, cas5_TypeI, cas6_TypeIE, cas7_TypeIE	Class 1	I	IE	-	DNA
*L. ellisii*	P1	cas3a_TypeI	Class 1	I	-	-	DNA
*L. gomenensis*	P1	Not detected	-	-	-	-	-
*L. kirschneri*	P1	cas1_TypeIC, cas2_TypeI-II-III, cas3_TypeI, cas4_TypeI-II, cas5_TypeIA, cas5c_TypeIC, cas7c_TypeIC, cas8c_TypeIC	Class 1	I, II, III	IC, IA	Cas5c, cas7c, cas8c	DNA
*L. interrogans*	P1	cas1_TypeIC, cas2_TypeI-II-III, cas3_TypeI, cas3a_TypeI, cas4_TypeI-II, cas5c_TypeIC, cas7c_TypeIC, cas8c_TypeIC	Class 1	I, II, III	IC	Cas3a	DNA
*L. kmetyi*	P1	Not detected	-	-	-	-	-
*L. mayottensis*	P1	cse1_TypeIE, cse2_TypeIE, cas1_TypeIE, cas2_TypeIE, cas3_TypeI, cas5_TypeIE, cas6_TypeIE, cas7_TypeIE	Class 1	I	IE	-	DNA
*L. noguchii*	P1	cas2_TypeI-II-III, cas3_TypeI, cas3a_TypeI, cas4_TypeI-II, cas5c_TypeIC, cas7c_TypeIC, cas8c_TypeIC	Class 1	I, II, III	IC	Cas3a, cas5c, cas7c,cas8c	DNA
*L. sanjuanensis*	P1	Not detected	-	-	-	-	-
*L. santarosai*	P1	cse1_TypeIE, cas1_TypeIE, cas1_TypeIA, cas2_TypeI-II-III, cas2_TypeIE, cas3_TypeI, cas5_TypeIE, cas5a2_TypeIA, cas6_TypeIE, cas6_TypeIA, cas7_TypeIE, cas7_TypeI, cas8a1a3_TypeIA	Class 1	I, II, III	IE, IA	Cas5a2, cas8a1a3	DNA
*L. stimsonii*	P1	cse1_TypeIE, cse2_TypeIE, cas1_TypeIE, cas2_TypeIE, cas3_TypeI, cas5_TypeIE, cas6_TypeIE, cas7_TypeIE	Class 1	I	IE	-	DNA
*L. tipperaryensis*	P1	Not detected	-	-	-	-	-
*L. weilii*	P1	cse1_TypeIE, cse2_TypeIE, cas1_TypeIE, cas2_TypeIE, cas3_TypeI, cas5_TypeIE, cas6_TypeIE, cas7_TypeIE	Class 1	I	IE	-	DNA
*L. yasudae*	P1	Not detected	-	-	-	-	-
*L. gorisiae*	P1	cse1_TypeIE, cpf1_TypeU (Cas12a), cas1_TypeIE, cas1_TypeU, cas2_TypeIE, cas2_Type I-II-III, cas3_TypeI, cas4_TypeU, cas5_TypeIE, cas6_TypeIE, cas7_TypeIE	Class 1 Class 2	II, V	IE	-	DNA
*L. andrefontaineae*	P2	cas3_TypeI	Class 1	I	-	-	DNA
*L. broomii*	P2	cas1_TypeIA, cas2_TypeI-II-III, cas3_TypeI, cas5a2_TypeIA, cas6_TypeIA, cas7_TypeI, cas8a1a3_TypeIA	Class 1	I, II, III	IA	Cas5a2, cas8a1a3	DNA
*L. dzoumogneensis*	P2	cas3_TypeI, cas3a_TypeI	Class 1	I	-	Cas3a	DNA
*L. fainei*	P2	cse1_TypeIE, cse2_TypeIE, cas1_TypeIE, cas2_TypeIE, cas3_TypeI, cas5_TypeIE, cas6_TypeIE, cas7_TypeIE	Class 1	I	IE	-	DNA
*L. fletcheri*	P2	cas1_TypeI-II-III, cas2_TypeI-II-III, cas4_TypeI-II, cas9_TypeIIB	Class 2	II	IIB	-	DNA
*L. fluminis*	P2	Not detected	-	-	-	-	-
*L. haakeii*	P2	csm2_TypeIIIA, cas3_TypeI	Class 1	I, III	IIIA	-	DNA and RNA
*L. hartskeerlii*	P2	cas3_TypeI	Class 1	I	-	-	DNA
*L. inadai*	P2	cas1_TypeIB, cas1_TypeU, cpf1_TypeU (Cas12a), cas1_TypeI-II-III, cas2_TypeI-II-III, cas3_TypeI, cas4_TypeI-II, cas4_TypeU, cas4_TypeI-II, cas5b_TypeIB, cas6_TypeI-III, cas7b_TypeIB	Class 1 Class 2	I, II, II, V	IB	Cas5b, cas7b	DNA
*L. johnsonii*	P2	Not detected	-	-	-	-	-
*L. koniambonensis*	P2	cse1_TypeIE, cse2_TypeIE, cas1_TypeIE, cas2_TypeIE, cas3_TypeI, cas5_TypeIE, cas6_TypeIE, cas7_TypeIE	Class 1	I	IE	-	DNA
*L. langatensis*	P2	Not detected	-	-	-	-	-
*L. licerasiae*	P2	cas3_TypeI	Class 1	I	-	-	DNA
*L. neocaledonica*	P2	Not detected	-	-	-	-	-
*L. perolatii*	P2	Not detected	-	-	-	-	-
*L. saintgironsiae*	P2	cas3_TypeI	Class 1	I	-	-	DNA
*L. sarikeiensis*	P2	Not detected	-	-	-	-	-
*L. selangorensis*	P2	Not detected	-	-	-	-	-
*L. semungkisensis*	P2	Not detected	-	-	-	-	-
*L. venezuelensis*	P2	Not detected	-	-	-	-	-
*L. wolffii*	P2	Not detected	-	-	-	-	-
*L. cinconiae*	P2	Not detected	-	-	-	-	-
*L. abararensis*	S1	cas3_TypeI, cas3a_TypeI	Class 1	I	-	Cas3a	DNA
*L. bandrabouensis*	S1	cas3_TypeI	Class 1	I	-	-	DNA
*L. biflexa*	S1	Not detected	-	-	-	-	-
*L. bourretii*	S1	Not detected	-	-	-	-	-
*L. bouyouniensis*	S1	cas3_TypeI	Class 1	I	-	-	DNA
*L. brenneri*	S1	Not detected	-	-	-	-	-
*L. chreensis*	S1	cas3a_TypeI	Class 1	I	-	-	DNA
*L. congkakensis*	S1	cas3a_TypeI	Class 1	I	-	-	DNA
*L. ellinghausenii*	S1	Not detected	-	-	-	-	-
*L. harrisiae*	S1	Not detected	-	-	-	-	-
*L. jelokensis*	S1	Not detected	-	-	-	-	-
*L. kanakyensis*	S1	Not detected	-	-	-	-	-
*L. kemamanensis*	S1	cas3_TypeI	Class 1	I	-	-	DNA
*L. levettii*	S1	Not detected	-	-	-	-	-
*L. meyeri*	S1	Not detected	-	-	-	-	-
*L. montravelensis*	S1	Not detected	-	-	-	-	-
*L. mtsangambouensis*	S1	Not detected	-	-	-	-	-
*L. noumeaensis*	S1	Not detected	-	-	-	-	-
*L. perdikensis*	S1	Not detected	-	-	-	-	-
*L. terpstrae*	S1	Not detected	-	-	-	-	-
*L. vanthielii*	S1	cas3a_TypeI	Class 1	I	-	-	DNA
*L. wolbachii*	S1	Not detected	-	-	-	-	-
*L. yanagawae*	S1	Not detected	-	-	-	-	-
*L. mgodei*	S1	Not detected	-	-	-	-	-
*L. milleri*	S1	Not detected	-	-	-	-	-
*L. iowaensis*	S1	Not detected	-	-	-	-	-
*L. paudalimensis*	S1	Not detected	-	-	-	-	-
*L. soteropolitanensis*	S1	Not detected	-	-	-	-	-
*L. limi*	S1	Not detected	-	-	-	-	-
*L. idonii*	S2	cas3_TypeI, cas3a_TypeI	Class 1	I	-	cas3a	DNA
*L. ilyithenensis*	S2	cas1_TypeU, cas2_TypeI-II-III, cas3_TypeI, cas4_TypeU, cpf1_TypeU(Cas12a)	Class 2	I, II, III, V	-	-	DNA
*L. kobayashii*	S2	Not detected	-	-	-	-	-
*L. ognonensis*	S2	Not detected	-	-	-	-	-
*L. ryugenii*	S2	cse1_TypeIE, cse2_TypeIE, cas1_TypeIE, cas2_TypeIE, cas3_TypeI, cas5_TypeIE, cas6_TypeIE, cas7_TypeIE	Class 1	I	IE	-	DNA

**Table 3 pathogens-14-01044-t003:** Distribution of Cas proteins across the three functional stages of the CRISPR-Cas system in the 19 *Leptospira* species encoding multiple proteins: Adaptation (adapter integration), Expression (pre-crRNA processing), and Interference (effector complex formation or target cleavage). Color coding indicates subgroup: green for pathogenic P1, blue for pathogenic P2, and red for saprophytic S2.

Species	Adaptation Spacer Integration	Expression Pre-crRNA Processing	Interference		Taxonomic Classification
			Effector Complex	Target Cleavage	
*L. adleri*	Cas1_TypeIA, Cas2_TypeI-II-III, Cas4_TypeI-II	Cas6_TypeIA	Cas5a2_TypeIA, Cas7b_TypeIB	Cas3_TypeI, Cas3a_TypeI	Class 1 type IF2
*L. alexanderi*	Cas1_TypeIE, Cas2_TypeIE, Cas4_TypeI-II	Cas6_TypeIE	Cas5_TypeIE, Cas7_TypeIE, Cse1_TypeIE, Cse2_TypeIE	Cas3_TypeI	Class 1, type IE
*L. alstonii*	Cas1_TypeIA, Cas2_TypeI-II-III, Cas4_TypeI-II	Cas6_TypeIE	Cas5a2_TypeIA, Cas5_TypeIE, Cas7_TypeI	Cas3_TypeI	Class 1 type IF2
*L. borgpetersenii*	Cas2_TypeIE	Cas6_TypeIE	Cas5_TypeI, Cas7_TypeIE, Cse1_TypeIE	Cas3_TypeI	Class 1, type IE
*L. gorisiae*	Cas1_TypeU, Cas1_TypeIE, Cas2_Type_I-II-III, Cas2_TypeIE, Cas4_TypeU	Cas6_TypeIE	Cas5_TypeIE, Cas7_TypeIE, Cse1_TypeIE	Cas12a Cas3_TypeI	Class 1, type IE Class 2 type V
*L. interrogans*	Cas1_TypeIC, Cas2_TypeI-II-III, Cas4_TypeI-II		Cas5c_TypeIC, Cas7c_TypeIC, Cas8c_TypeIC	Cas3_TypeI, Cas3a_TypeI	Class 1 type IC
*L. kirschneri*	Cas1_TypeIC, Cas2_TypeI-II-III, Cas4_TypeI-II		Cas5_TypeIA, Cas5c_TypeIC, Cas7c_TypeIC, Cas8c_TypeIC	Cas3_TypeI	Class 1 type IC
*L. mayottensis*	Cas1_TypeIE, Cas2_TypeIE	Cas6_TypeIE	Cas5_TypeIE, Cas7_TypeIE, Cse1_TypeIE, Cse2_TypeIE	Cas3_TypeI	Class 1, type IE
*L. noguchii*	Cas2_TypeI-II-III, Cas4_TypeI-II		Cas5c_TypeIC, Cas7c_TypeIC, Cas8c_TypeIC	Cas3_TypeI, Cas3a_TypeI	Class 1 type IC
*L. santarosai*	Cas1_TypeIA, Cas1_TypeIE, Cas2_TypeI-II-III, Cas2_TypeIE	Cas6_TypeIA, Cas6_TypeIE	Cas5_TypeIE, Cas5a2_TypeIA, Cas7_TypeIE, Cas7_TypeI, Cas8a1a3_TypeIA, Cse1_TypeIE	Cas3_TypeI	Class 1 type B
*L. stimsonii*	Cas1_TypeIE, Cas2_TypeIE	Cas6_TypeIE	Cas5_TypeIE, Cas7_TypeIE, Cse1_TypeIE, Cse2_TypeIE	Cas3_TypeI	Class 1, type IE
*L. weilii*	Cas1_TypeIE, Cas2_TypeIE	Cas6_TypeIE	Cas5_TypeIE, Cas7_TypeIE, Cse1_TypeIE, Cse2_TypeIE	Cas3_TypeI	Class 1, type IE
*L. broomii*	Cas1_TypeIA, Cas2_TypeI-II-III	Cas6_TypeIA	Cas5a2_TypeIA, Cas7_TypeI, Cas8a1a3_TypeIA	Cas3_TypeI	Class 1 type B
*L. fainei*	Cas1_TypeIE, cas2_TypeIE	Cas6_TypeIE	Cas5_TypeIE, Cas7_TypeIE, Cse1_TypeIE, Cse2_TypeIE	Cas3_TypeI	Class 1, type IE
*L. fletcheri*	Cas1_TypeI-II-III, Cas2_TypeI-II-III, Cas4_TypeI-II	RNAse III		Cas9_TypeIIB	Class 2 type IIB
*L. inadai*	Cas1_TypeIB, Cas1_TypeU, Cas1_TypeI-II-III, Cas2_TypeI-II-III, Cas4_TypeI-II, Cas44_TypeU, Cas4_TypeI-II	Cas6_TypeI-III	Cas5b_TypeIB, Cas7b_TypeIB	Cas3_TypeI, Cas12a	Class 1 type IF2 Class 2 type V
*L. koniambonensis*	Cas1_TypeIE, Cas2_TypeIE	Cas6_TypeIE	Cas5_TypeIE, Cas7_TypeIE, Cse1_TypeIE, Cse2_TypeIE	Cas3_TypeI	Class 1, type IE
*L. ilyithenensis*	Cas1_TypeU, Cas2_TypeI-II-III, Cas4_TypeU			Cas3_TypeI, Cas12a	Class 2 type V
*L. ryugenii*	Cas1_TypeIE, Cas2_TypeIE	Cas6_TypeIE	Cas5_TypeIE, Cas7_TypeIE, Cse1_TypeIE, Cse2_TypeIE	Cas3_TypeI	Class 1, type IE

## Data Availability

Data are contained within the article (Appendix B).

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
