# Peer review of "Bioinformatic Identification of CRISPR–Cas Systems in Leptospira Genus: An Update on Their Distribution Across 77 Species"

_pathogens, 2025, doi:10.3390/pathogens14101044_

Round 1

Reviewer 1 Report

Comments and Suggestions for Authors

In this work, the authors provide a paper that appears to show a comprehensive bioinformatic survey of CRISPR-Cas systems across 77 Leptospira species, revealing the presence of diverse Cas proteins, direct repeats, spacers, and prophages. While the proposed methodology is promising and the experiments are solid done, it is apparent that major revisions should be completed prior to publication.

  1. This manuscript uses RAST and CRISPRCasFinder for Cas gene and CRISPR array detection. Could you clarify whether results from both tools were cross-validated? Were discrepancies observed between them, and if so, how were they resolved?
  2. The study relies heavily on automated annotation pipelines (Prokka, RAST, InterProScan, PHASTER). Please describe the criteria used to confirm that detected Cas genes are intact (i.e., not pseudogenes or truncated ORFs).
  3. Are there differences in domain architecture or sequence motifs of Leptospira Cas9/Cas12a compared to model species (e.g., Streptococcus pyogenes Cas9, Francisella novicida Cas12a)?
  4. Some species carry two CRISPR-Cas systems (e.g., L. inadai). How do you interpret this?
  5. You conclude that CRISPR-Cas may serve as a genome editing tool in Leptospira. While this is exciting, the current evidence is limited to in silico prediction. Could you give some discussion about the distinguish between bioinformatic detection of Cas proteins and experimental validation of editing capability?

Author Response

Comments and Suggestions for Authors

In this work, the authors provide a paper that appears to show a comprehensive bioinformatic survey of CRISPR-Cas systems across 77 Leptospira species, revealing the presence of diverse Cas proteins, direct repeats, spacers, and prophages. While the proposed methodology is promising and the experiments are solid done, it is apparent that major revisions should be completed prior to publication.

  1. This manuscript uses RAST and CRISPRCasFinder for Cas gene and CRISPR array detection. Could you clarify whether results from both tools were cross-validated? Were discrepancies observed between them, and if so, how were they resolved?

Thank you very much for your input and your concern about cross-validation. We are happy to clarify how this process was done. The results from RAST and CRISPRCasFinder were combined into a single database. Duplicates were removed, while unique sequences from each tool were retained. For the sequences that could not be associated to any phage between the two tools, we kept the counts reported by CRISPRCasFinder, assuming that the lower counts from RAST were included within those identified by CRISPRCasFinder. The following table shows the total number of proteins, spacers, and repeats found in each server. However, please keep in mind that these numbers were only reported for the species with complete functional arrays.

RAST

CRISPRCasFinder

Proteins

132

184

Spacers

495

723

Repeats

523

928

  1. The study relies heavily on automated annotation pipelines (Prokka, RAST, InterProScan, PHASTER). Please describe the criteria used to confirm that detected Cas genes are intact (i.e., not pseudogenes or truncated ORFs).

Initially, the 77 genomes used were fully annotated, and pseudogenes were detected. These results can be seen in column 9 of Table 1 (Non-coding genes) using the bioinformatics programs Pseudopipe, Pseudofinder, and RetroScan. Additionally, it was verified that the genes encoding the CAS proteins were not included in the list of non-coding genes.

The following paragraph is added (Lines 133-140):

The genomes were organized by species subgroups, and the genomic characteristics were identified (species names, clade, accession number, genome size (Mb), GC content (%), genes, proteins, and non-coding sequence) using the bioinformatics program Prokka V1.14.5 (https://github.com/tseemann/prokka), Pseudopipe (https://github.com/mosilab/pseudopipe) , Pseudofinder (https://github.com/filip-husnik/pseudofinder), and RetroScan (https://github.com/Vicky123wzy/RetroScan) [49]. Genomic characteristics were compared between different species to determine the average, highest and lowest values.

  1. Are there differences in domain architecture or sequence motifs of Leptospira Cas9/Cas12a compared to model species (e.g., Streptococcus pyogenes Cas9, Francisella novicida Cas12a)?

When comparing the Cas9 and Cas12a proteins of the Leptospira genus with species that have served as models for the study of the Crispr-Cas system (Streptococcus pyogenes Cas9, Francisella novicida Cas12a), the following results were obtained: Cas9 y Cas12a are very similar to those proteins found in other species of the Leptospira genus. Furthermore, they have a high percentage of identity, genetic relatedness, and coverage with genetically related genera such as Turnerella and Leptonema.

There are differences between the Cas9 protein of Leptospira fletcheri and Strepto-coccus pyogenes in the amino acid sequence, functional domains, and proteins structure. However, the Cas12a protein of Leptospira inadai contains sequence, functional domains, and structure identical to those found in Francisella novicida, which is a model species in the study of the CRISPR-Cas system (supplementary material S2).

Supplementary material S2 is added to show these results. the change in the text on the lines (502-512) is highlighted.

  1. Some species carry two CRISPR-Cas systems (e.g., L. inadai). How do you interpret this?

The following paragraph was added to the publication (lines 690-700):

Bacterial species can harbor and activate multiple types of CRISPR-Cas systems, allowing them to defend against a wider range of viral and genetic threats. The presence of multiple systems reflects an evolutionary war with phages, which in turn have developed inhibitors such as anti-CRISPR proteins to evade detection1. Besides, an additional system can act as a backup in case the primary CRISPR-Cas system fails, ensuring the continuity of bacterial defense. The presence of multiple CRISPR-Cas systems in a single bacterial species is a known phenomenon: some Streptococcus species can harbor up to three different types of CRISPR-Cas systems2. Clostridium and Bacillus are genera that have also shown the presence of multiple types of CRISPR-Cas systems in different strains, such as types IB, II-B, II-C, and III-B in Clostridium3, and types IB and IC in Bacillus4.

  • Zhang, F., Song, G., & Tian, Y. (2019). Anti-CRISPRs: The natural inhibitors for CRISPR-Cas systems. Animal models and experimental medicine, 2(2), 69–75. https://doi.org/10.1002/ame2.12069

  • Carte, J., Christopher, R. T., Smith, J. T., Olson, S., Barrangou, R., Moineau, S., Glover, C. V., 3rd, Graveley, B. R., Terns, R. M., & Terns, M. P. (2014). The three major types of CRISPR-Cas systems function independently in CRISPR RNA biogenesis in Streptococcus thermophilus. Molecular microbiology93(1), 98–112. https://doi.org/10.1111/mmi.12644.

  • McAllister, K. N., & Sorg, J. A. (2019). CRISPR Genome Editing Systems in the Genus Clostridium: a Timely Advancement. Journal of bacteriology201(16), e00219-19. https://doi.org/10.1128/JB.00219-19

  • Naiymeh Sheykholeslami, Hamid Mirzaei, Yousef Nami, Jalil Khandaghi, Afshin Javadi, Ecological and evolutionary dynamics of CRISPR-Cas systems in Clostridium botulinum: Insights from genome mining and comparative analysis, Infection, Genetics and Evolution, Volume 123, 2024, 105638, ISSN 1567-1348, https://doi.org/10.1016/j.meegid.2024.105638.

  1. You conclude that CRISPR-Cas may serve as a genome editing tool in Leptospira. While this is exciting, the current evidence is limited to in silico prediction. Could you give some discussion about the distinguish between bioinformatic detection of Cas proteins and experimental validation of editing capability?

The text clarifies that the bioinformatics analysis of CRISPR-Cas systems in the Leptospira genus is a preliminary result that demonstrates the existence of the components, but it is necessary to experimentally validate these results to ensure that the systems work correctly, for example: evaluate experimentally the degradation of a bacteriophage or block a gene in a Leptospira specie that has the complete CRISPR-Cas system.

The following paragraph is added to the paper (lines 728-732):

The bioinformatics analysis of CRISPR-Cas systems in the Leptospira genus is a preliminary result that demonstrates the existence of the components, but it is necessary to experimentally validate these results to ensure that the systems work correctly, for example: evaluate experimentally the degradation of a bacteriophage or block a gene in a Leptospira species that has the complete CRISPR-Cas system.

Thank you very much for the corrections, they helped improve the document!!!!!!!!!!!

Reviewer 2 Report

Comments and Suggestions for Authors

Leptospira genome analysis revealed Cas proteins in 36 out of 77 species, including Cas1–Cas9 and Cas12, representing class 1 and 2 CRISPR-Cas systems (types I, II, and V). Direct repeats and spacers were found in 19 species, with conserved nucleotide motifs and matches to 323 bacteriophages. Three intact bacteriophages were detected in four Leptospira genomes. These findings suggest that functional CRISPR-Cas systems exist in at least 19 Leptospira species, potentially conferring phage defense.

  1. Does the Cas3 proteins found in some species have other biological roles beyond CRISPR-Cas defense?
  2. Why only 19 out of 77 Leptospira species have a complete functional Cas system
  3. How are the Cas9 and Cas12 proteins in Leptospira structurally or functionally distinct from others
  4. How can large numbers of hypothetical proteins found in Leptospira genomes be validated

Author Response

Comments and Suggestions for Authors

Leptospira genome analysis revealed Cas proteins in 36 out of 77 species, including Cas1–Cas9 and Cas12, representing class 1 and 2 CRISPR-Cas systems (types I, II, and V). Direct repeats and spacers were found in 19 species, with conserved nucleotide motifs and matches to 323 bacteriophages. Three intact bacteriophages were detected in four Leptospira genomes. These findings suggest that functional CRISPR-Cas systems exist in at least 19 Leptospira species, potentially conferring phage defense.

  1. Does the Cas3 proteins found in some species have other biological roles beyond CRISPR-Cas defense?

The following paragraph was added to the manuscript (lines 573-580):

Beyond defense against viruses (CRISPR immunity), the Cas3 protein may be involved in regulating biofilm formation and virulence in bacteria [1], as well as influencing genes related to quorum sensing (a form of bacterial communication where cells release and detect signal molecules called autoinducers, allowing them to sense the population density of their community and coordinate gene expression to alter collective behavior) [2], secretion systems [3], and flagella formation [4]. These non-canonical functions of Cas3 are being investigated and could have implications for gene editing and other biological processes in cells.

  • Guo, T., Yang, J., Zhou, N. et al.Cas3 of type I-Fa CRISPR-Cas system upregulates bacterial biofilm formation and virulence in Acinetobacter baumanniiCommun Biol 8, 750 (2025). https://doi.org/10.1038/s42003-025-08124-6
  • Cui, L., Wang, X., Huang, D., Zhao, Y., Feng, J., Lu, Q., Pu, Q., Wang, Y., Cheng, G., Wu, M., & Dai, M. (2020). CRISPR-cas3of Salmonella Upregulates Bacterial Biofilm Formation and Virulence to Host Cells by Targeting Quorum-Sensing Systems. Pathogens9(1), 53. https://doi.org/10.3390/pathogens9010053
  • He, L., St. John James, M., Radovcic, M., Ivancic-Bace, I., & Bolt, E. L. (2020). Cas3 Protein—A Review of a Multi-Tasking Machine. Genes11(2), 208. https://doi.org/10.3390/genes11020208.
  • Guo, T., Yang, J., Zhou, N., Sun, X., Huan, C., Lin, T., Bao, G., Hu, J., & Li, G. (2025). Cas3 of type I-Fa CRISPR-Cas system upregulates bacterial biofilm formation and virulence in Acinetobacter baumannii. Communications biology, 8(1), 750. https://doi.org/10.1038/s42003-025-08124-6
  1. Why only 19 out of 77 Leptospira species have a complete functional Cas system

The following paragraph was added to the manuscript (lines 736-746):

Some bacterial species lack CRISPR-Cas systems due to evolutionary inactivation; CRISPR-Cas systems can be selectively inactivated in certain bacteria, as observed in the Bacillus cereus group [1], to adapt to specific environments. Another cause of CRISPR-Cas system loss is selective pressure induced by antibiotic exposure. Intensive antibiotic use can lead to the loss of the CRISPR-Cas system, as it can act as a barrier to the acquisition of antibiotic resistance genes [2]. Additionally, total or partial loss of essential components of the CRISPR-Cas system can render the system inoperable. For these reasons, we assume that some species of the Leptospira genus have lost their CRISPR-Cas defense system.

  • Zheng, Z., Zhang, Y., Liu, Z. et al. The CRISPR-Cas systems were selectively inactivated during evolution of Bacillus cereus group for adaptation to diverse environments. ISME J 14, 1479–1493 (2020). https://doi.org/10.1038/s41396-020-0623-5.

  • Arias, C. A., & Murray, B. E. (2012). The rise of the Enterococcus: beyond vancomycin resistance. Nature reviews. Microbiology10(4), 266–278. https://doi.org/10.1038/nrmicro276.

  1. How are the Cas9 and Cas12 proteins in Leptospira structurally or functionally distinct from others

These proteins are very similar to those found in other species of the Leptospira genus. Furthermore, they have a high percentage of identity, genetic relatedness, and coverage with genetically related genera such as Turnerella and Leptonema.

There are differences between the Cas9 protein of Leptospira fletcheri and Strepto-coccus pyogenes in the amino acid sequence, functional domains, and proteins structure. However, the Cas12a protein of Leptospira inadai contains sequence, functional domains, and structure identical to those found in Francisella novicida, which is a model species in the study of the CRISPR-Cas system (supplementary material S2).

Supplementary Table S2 is added to show these results. the change in the text on the lines (502-512) is highlighted.

  1. How can large numbers of hypothetical proteins found in Leptospira genomes be validated

Validation of hypothetical proteins in the Leptospira genus can be performed by bioinformatics analysis and experimental assays.

Bioinformatics analysis in hypothetical protein of the Leptospira genus, can be used to evaluate the amino acid sequence and presence of mutations. Predicting its three-dimensional structure can determine its shape and identify functional domains. Functional domain analysis can determine its biological function, the biological processes in which it participates, and cellular location. Analyzing its physicochemical properties can determine molecular weight, isoelectric point, amino acid composition, properties of radical groups, atomic composition, extinction coefficient, half-life, instability index, the presence of a signal peptide and cellular location, interaction with other proteins, location in a metabolic pathway, and its possible association with known diseases.

In parallel, experimental analyses are performed, such as gene sequencing and its location in the genome, transcriptomic analysis to validate its transcription level, and proteomic analysis to validate protein translation. Additionally, the protein can be sequenced using mass spectrometry to ensure its identification. Finally, in vitro and in vivo functional analyses are performed to assign a final biological function. For example: evaluate experimentally the degradation of a bacteriophage or block a gene in a Leptospira specie that has the complete CRISPR-Cas system.

The following paragraph was added to the text to clarify the difference between a bioinformatic prediction and an experimental validation test (lines 717-721 are highlighted in blue).

The following publication presents the biological characterization of hypothetical proteins in the Leptospira genus.

Bidkar, A., Thakur, N., Bolshette, J. D., & Gogoi, R. (2014). In-silico Structural and Functional analysis of Hypothetical proteins of Leptospira Interrogans. Biochem Pharmacol3(136), 2167-0501.

Thank you very much for the corrections, they helped improve the document!!!!!!!!!!!

Round 2

Reviewer 1 Report

Comments and Suggestions for Authors

I think the author addressed all my concerns and it can be published.